# Characterisation of Wheat Straw Pellets Individually and in Combination with Cassava Starch or Calcium Carbonate under Various Compaction Conditions: Determination of Pellet Strength and Water Absorption Capacity

**DOI:** 10.3390/ma13194375

**Published:** 2020-10-01

**Authors:** Patryk Matkowski, Aleksander Lisowski, Adam Świętochowski

**Affiliations:** Department of Biosystems Engineering, Institute of Mechanical Engineering, Warsaw University of Life Sciences, Nowoursynowska 166, 02-787 Warsaw, Poland; patryk_matkowski@sggw.edu.pl (P.M.); adam_swietochowski@sggw.edu.pl (A.Ś.)

**Keywords:** additives, crushed pellets, optimization, regression models, tensile strength

## Abstract

This study aimed to optimise the production conditions of wheat straw (WS) pellets and pellets with the additives of cassava starch (CS) or calcium carbonate (CC) based on the criteria of pellet strength and water absorption by crushed pellets. The pellets produced using a 2–10%-wt/wt additive ratio, material moisture of 10–30% w.b., die height of 66–86 mm, and material temperature of 78–108 °C were tested. The influence these factors on the strength parameters of pellets was different than on the water absorption by the crushed pellets. The pellets made of WS blended with CC additive were characterised by better strength parameters and the compressed pellets were characterised by better water absorption than those with CS. High and positive correlation among specific pellet compression work, elasticity modulus for pellet compression, and tensile strength values were observed. As the strength parameters of pellets showed high correlation with single pellet density, for the consistency of conclusions, the optimal conditions for pellet production were assumed based on the density. For optimal conditions at 4% wt/wt additive ratio, 23% w.b. material moisture, 78 mm die height, and 80 °C material temperature, the specific pellet compression work was 3.22 mJ·mm^−2^, elasticity modulus was 5.78 MPa, and maximum tensile strength of the pellets was 2.68 MPa; moreover, the water absorption by crushed pellets amounted to 2.60 g H_2_O·g^−1^ of dry matter.

## 1. Introduction

The bedding material type, particles size, as well as moisture bedding and caking, have been identified as the main factors to birds’ welfare [1]. In comparative tests of pelleted straw, chopped wheat straw (WS), wood chips, rice straw, and shredded paper as bedding material, pelleted straw showed the lowest incidence of foot-pad dermatitis (FPD) [2]. The occurrence or appearance of FPD is less frequent by 24 to 29 days among birds kept on pelleted straw in compared to birds kept on chopped straw and paper. This may be connected with its ability to absorb water and vaporisation speed as pelleted straw is denser than chopped straw, causing less crushing of the bedding material and creation of small particles. The results of comparative studies of four classes of western red cedar (*Thuja plicata*) wood chips and Douglas fir (*Pseudotsuga menziesii*) with western juniper (*Juniperus occidentalis*) show that the absorption capacity and rate of moisture release increase with decreasing particle size [3]. Smaller bedding particle sizes and increased bedding quality are associated with reduced FPD [4].

Sharp edges of chopped straw may be responsible for the occurrence of FPD [1]. Bedding material with large particle size and high moisture content also had an impact on the development of FPD [4]. The significantly high correlation between high bedding material moisture, caking results, and FPD occurrence frequency has been previously recorded [1]. Chipped pine, chopped straw, cotton-gin trash, and pine shaving bedding have shown the highest correlation. Mortar sand and ground door filler (a wood fibre-based material used in insulating metal doors) showed the lowest value of FDP occurrence. The ability of bedding to absorb and quickly release moisture may be the most important bedding feature for preventing FDP [1].

The evaluation of pellet quality can be conducted using various indicators. Previous considerations that take into account the relationships among parameters indicate the effect of pellet production, with its ability to absorb water, is important and significant to this study [2,5]. These parameters are also relevant to energy pellets, where their ability to absorb water or moisture should be as low as possible. Water absorption by biomass is the process of resulting molecules, atoms, or ions and adsorption is a surface phenomenon. Often these phenomena occur simultaneously and are referred to as sorption. The absorption mechanism of bedding or fluid-treated pellets by the entire volume of particles dominates and whose intensity is time, is defined as the diffusion coefficient. Moisture diffusion in the material structure depends on the type of the material, volume of micro gaps among particles, and volume of particles or pellets, fibre fraction volumes, the speed of the mass stream, time, additives, moisture, relative humidity, and air temperature [6,7]. Smaller particles and higher temperature increase the speed of absorption. The modification of the structure of the pellets intended for the bedding should be directed on enhancing hygroscopic characteristics, contrary to energy pellets. There are more moisture absorption data results from the air by energy pellets made of blends of various materials, than by the pellets intended for bedding. If the fuel pellets do not meet the trade conditions, they can be used for bedding [5].

The moisture content of crushed straw was statistically higher than pelleted wheat straw, chipwood, and rice straw [2,8]. This result may be linked with the higher ability of crushed straw to absorb and release the water.

The comparative studies with pine shavings and sawdust have shown that shredded pellets have a good moisture retention capacity, an acceptable level of fine particles, and a low level of chemical contamination [9].

The production of stable pellets with high density requires the optimal range of biomass moisture. In the defined temperature, water acts as a bonding agent, enhancing the bonds among material particles, helping to develop van der Waals forces, and increasing the area of contact among moister particles [10]. The lignin melting point can be lowered from 140 °C [11] to 100–135 °C (usually achieved during commercial pelleting) using moisture of 8–15% in biomass raw materials [12].

The analysis of several factors influencing the quality of pellets, especially density and strength, requires optimisation methods. For this purpose, linear and nonlinear regression equations are used, as well as more advanced analytic methods. The optimisation is based on linking all these aims with respect to the relevant factors [13].

There are not enough research results in the available literature for bedding pellets made of WS with cassava starch (CS) or calcium carbonate (CC) additives at various moisture and density parameters. There is also not enough information about the optimal conditions for undertaking the process of densification of such blends due to the strength of pellets and water absorption ability by crushed pellets intended for bedding.

The aim of this study was to determine the most favourable combinations of the density parameters of WS by itself and blended with CS or CC as bonding additives by maximising the pellets’ compressive strength and water absorption by crushed pellets (*k*). The parameters characterising the pellets compressive strength were specific pellet compression work (*E_j_*), during which the pellets cracked; elasticity modulus (*E*); and maximum tensile strength (*σ_c_*). Crushed pellets as a potential material for bedding were accessed on the basis of *k*. In the analysis of these research results, the single pellet density (*ρ_p_*) was also taken into account in the evaluation of the compression process. The control variables were the participation of bonding additives CC and CS, material moisture content (*MC*), die height (*l_c_*), and material temperature (*t*) during agglomeration pressure. The obtained data results, being criterion indicators for the quality of pellets, were elaborated in a statistical manner using variant analysis and features correlation. For the response function, mathematical models and optimisation of compaction conditions due to assumed criterion indicators were developed. The novelty in the work was the type of logically selected bonding additives to WS in the production of pellets for bedding and its influence on the quality of pellets, including on *k* and the optimisation of compaction conditions using Scilab v.6.02 software.

The selection of additives and factors with their level of values were presented in our two previous articles [14,15].

## 2. Materials and Methods

### 2.1. Materials

For this study, the pellets made of WS were used with CC or CS as bonding additives, with the involvement of three additive ratios (*A*) (2, 6, and 10% wt/wt) and three levels of *MC* (10, 20, and 30% w.b.). The pellets were made of die in an open chamber with adjustable heights of 66, 76, and 86 mm at temperatures of 78, 93, and 108 °C (Table 1). The additive ratio of 0% wt/wt means pure WS. These described factors are the most important characteristics of the material and the pelleting process. The range of values for these factors was selected on the basis of preliminary research with taking into account the practical use. More information about the selection of factors were provided in our first article [15]. The selection of additives was dictated by their physicochemical properties, hygroscopicity, neutrality towards the environment and the positive effect on the litter properties. A wide range of factor values will allow for more reliable determination of optimal pelleting conditions, due to the strength of the pellets and their water absorption capacity. In order to receive the more reliable database, the tests for each combination of factors and levels were performed.

### 2.2. Material Moisture Content (MC)

Immediately before the measurements of pellet compression and *k*, the *MC* was determined [16]. The samples of the milled material with a weight of 2 g each were weighed on an analytical balance (WPA 40/160/C/1, Radwag, Radom, Poland) with a 0.00001 g accuracy and dried at a temperature of 105 ± 2 °C for 24 h using an SLW 115 laboratory dryer (Pol-Eko Aparatura, Wodzisław Śląski, Poland). The moisture measurements were reported three times for every combination. As the samples were stored in a room with the same temperature of 20 ± 2 °C and relative humidity of 68% ± 5%, they achieved moisture equilibrium moisture. The average value of the *MC* of all combinations was estimated as 8.15% ± 0.03% w.b. The research results were converted into a dry matter (DM).

### 2.3. Pellets Strength to Compressive Loads

The strength of single pellets for compressive loads was determined by diametric compression tests, which have been widely used by researchers [17,18,19]. The pellets were cut in 20 mm pieces using a scalpel. The front surfaces of the pellets were ground and the samples placed in triangular element to keep the surfaces perpendicular to the pellet axis. The diameter of pellets was measured at half of their length in two perpendicular directions. Their length was measured using a MAUa-E2 4F digital calliper (FK Vis, Warsaw, Poland) with an accuracy of 0.01 mm. Each pellet was weighed on an electronic scale (WPS 600/C, Radwag, Radom, Poland) with an accuracy of 0.01 g.

The pellets were placed individually on the steel plate of the TIRAtest (Matest, Łódź, Poland) universal testing machine and were compressed with a 25 × 50 mm^2^ punch with a speed of 5 mm·min^−1^ until the moment when the piston shifted 4 mm (8 mm pellet diameter) (Appendix A). The measurements were performed three times for each combination of the pellet production. The pellets compressive strength was assessed by determining *E_j_*, *E*, and *σ_c_* [20]. As this parameter is defined differently in the literature, the term more commonly used has been adopted [21].

### 2.4. Pellet Strength Parameters during Compression

Based on the set of data force-displacement, the strength parameters during compression were set as *E_j_*, *E*, and *σ_c_* at which the pellet cracked based on the following relationships:(1)$$E_{j}=\frac{1}{S}\int_{0}^{\Delta l}Fdx$$
(2)$$E=\frac{Fd}{S\Delta l}$$
(3)$$\sigma_{c}=\frac{2F_{max}}{\pi dl_{p}},$$
where *E_j_* is the specific compression work, in mJ·mm^−2^, until pellet cracking is achieved; *F* is the compression force in N; *S* is the surface of the pellet subjected to load in mm^2^; *x* is the punch displacement in mm, equal to pellet deformation; *E* is the elasticity modulus at compression in MPa; *d* is the pellet diameter in mm before loading; Δ*l* is the radial displacement of the punch (pellet deformation under load) in mm; *σ_c_* is the maximum tensile strength during pellet cracking in MPa; *F*_max_ is the maximum compression force in N; and *l_p_* is the pellet length in mm.

### 2.5. Water Absorption

Directly after the pellet compression tests, the samples of the crushed pellets were placed in a coffee brewer of know weight (tare) and weighed on an electronic scale (WLC1/A2, Radwag, Radom, Poland) with an accuracy ±0.01 g. These tests were conducted by immersing the samples in purified water at 20 ± 1 °C for 30 min. The coffee brewers with pellets were left to drain for 5 min. After taking them out of the water, they were gently shaken three times to remove excess water from the screen of the sieve and weighed again (Appendix A).

The difference in mass of the sample after and before immersing is *k*. The value of *k*, based on dry matter mass, was calculated according to the following formula:(4)$$k=\left(m_{s1}-m_{s0}\right)/m_{s0}\left(1-MC/100\right),$$
where *k* is water absorption by crushed pellets in g H_2_O·g^−1^ DM; m*_s_*_1_ is the weight in g of the sample of pellets crushed after immersion in water; *m_s_*_0_ is a mass in g of sample of pellets crushed before immersion in water; and *MC* is the moisture in % of the pellets crushed before immersing in water

In the same way, for contrast, the water absorption of chopped WS and blends of WS with CS or CC with *A* values of 2%, 6%, and 10% wt/wt was tested. In the comparative research, only the material with the lowest moisture 10% w.b. was used because only dry material was used as bedding.

### 2.6. A Method for Optimising Compaction Conditions Due to the Pellets’ Strength and Their Water Absorption Capacity

The control variables for WS with CS or CC were: *A, MC*, *l_c_*, and *t*. The responses of the dependent variables, *E_j_*, *E*, *σ_c_*, and *k*, were calculated based on values obtained from experiments and can be defined as a general regression Equation:(5)$$y_{n}=\beta_{0}+\overset{4}{\underset{i=1}{\sum }}\beta_{i}x_{i}+\overset{4}{\underset{i=1}{\sum }}\beta_{ii}x_{i}^{2}+\overset{3}{\underset{i=1}{\sum }}\overset{4}{\underset{j=i+1}{\sum }}\beta_{ij}x_{i}x_{j},\;\left(n=1,2,3,4\right),$$
where *y*_1_ is the specific pellet compression work (*E_j_* in mJ·mm^−2^); *y*_2_ is elasticity modulus (*E* in MPa); *y*_3_ is a tensile strength of pellets (*σ_c_* in MPa); *y*_4_ is water absorption by crushed pellets (*k* in g H_2_O·g^−1^ DM); *x*_1_ is the additive ratio (*A* in % wt/wt); *x*_2_ is the material moisture content (*MC* in % w.b.); *x*_3_ is the die height (*l_c_* in mm); *x*_4_ is the material temperature (*t* in °C); *β*_0_, *β*_i_, *β*_ii_, and *β*_ij_ are the coefficients of intercept terms, linear terms, quadratic terms, and interaction terms in the equation, respectively.

For the objective function (Equation (5)) relative to the four control variables *A, MC*, *l_c_*, and *t*, the gradient *g_n_* was determined using finite differences in the matrix equation form.
(6)$$g_{n}=\left[\frac{\partial g_{n}}{\partial A},\;\frac{\partial g_{n}}{\partial MC},\;\frac{\partial g_{n}}{\partial l_{c}},\;\frac{\partial g_{n}}{\partial t}\right]$$

The minimising function, *y_n_*(*A*,*MC*,*l_c_*,*t*) was calculated at the initial conditions of the decision variables *A*_0_, *MC*_0_, *l_c_*_0_, and *t*_0_ being 4% wt/wt, 15% w.b., 70 mm, 90 °C, respectively, with the bound constraints being 0% wt/wt ≤ *A* ≤ 10% wt/wt, 10% w.b. ≤ *MC* ≤ 30% w.b., 66 mm ≤ *l_c_* ≤ 86 mm, and 78 °C ≤ *t* ≤ 108 °C.

To minimise the non-linear function $\underset{MC,l_{c},t,A}{\min}y_{n}\left(MC,\;l_{c},\;t,A\right)$, an “optim” function with the Quasi-Newton algorithm was used.

The optimal process parameters for the objective function and graphs were developed using the open-source software package Scilab v.6.0.2.

### 2.7. Statistical Analysis

The data were analysed for the influence of factors (the type of material, *A*, *MC*, *l_c_*, and *t*) on the resistance features of pellets (*E_j_*, *E*, *σ_c_*) and *k* using the multi-criteria MANOVA variance analysis with an *F* test (Fisher-Snedecor). The statistical importance of the differences between the average values of the parameters was determined using the Tukey test method in relation to criterion parameters characterising pellet strength and *k*. The analysis was conducted with an assumed significance level of *p* ≤ 0.05. Statistica v.13.3 (StatSoft Polska Ltd., Cracow, Poland) was used for the statistical analysis.

## 3. Results and Discussion

### 3.1. Research Results

In majority of the cases, the main factors (additive type, *A*, *MC*, *l_c_*, and *t*) of pellet quality parameters (*p* < 0.0002, Table 2) showed highly significant statistical differences.

Factor *A* was insignificant for *E_j_* (*p* = 0.0523), *E* (*p* = 0.5438), and *σ_c_* (*p* = 0.2428). The feature *l_c_* did not significantly affect the differences in the *E* (*p* = 0.4100). The matrix of correlation coefficients among the quality parameters of pellets and control variables is presented in Table. The *ρ_p_* value was included in the joint interpretation of the strength parameters and *k*.

The values of the regression coefficients for the mathematical models of strength parameters and *k* are shown in Appendix A. Their graphic interpretation is shown in Figure 1, Figure 2, Figure 3 and Figure 4. The optimal values of control variables, for which the objective functions characterising the quality of pellets reached their optimum tested limits of control variables, are presented in Table 3.

### 3.2. Specific Pellet Compression Work E_j_

The pellets made of the WS blend with CC were characterised by the largest *E_j_* (2.66 mJ·mm^−2^) during the deformation until cracking. The pellets made of pure WS required slightly less destructive work (by 5%) but without a statistically significant difference. The pellets made of the WS blend with CS were characterised by the lowest of *E_j_* (1.28 mJ·mm^−2^).

Cracking cross-sections depend on the particle size arrangement [22].and amount of adhesive bonds associated with surface wettability [23]. There is a positive relationship between wettability and wood adhesion [24].

The CS additive probably reduced the wettability of the surface and reduced the adhesion among particles whose grains were crystalline and finely porous [25]. CC grains form clusters that combine randomly with larger aggregates and have the ability to combine with lignocellulose particles, which makes the bonds durable [26].

Increasing the additive ratio to WS contributed to an insignificant decrease of *E_j_*. The CS additive decided of this averaged, negative downward trend of decreasing work vs. additive ratio. It can only be pointed out the tendency, because the statistical analysis shows that the influence of involvement of additives was not unambiguous (*p* = 0.0523), and the values of correlation coefficients were not correlated (Appendix A).

The pellets with increased moisture content were characterised by higher resistance to compression loads. The growth of *E_j_* was significantly higher in the first moisture range (10–20% w.b.) than in the second (20–30% w.b.), amounting to 124% and 2%, respectively.

In the second range, the differences were statistically insignificant and at the moisture level slightly higher than 20% w.b. the parameter reached the stabilisation state. The significant growth of *E_j_* was recorded at the highest *t* of 108 °C. At 78 °C and 93 °C, the change courses of *E_j_* reached plateaus (Figure 1). Contrary to the *MC*, the growth of the *t* decreased *σ_c_*. At ranges of 78–93 °C and 93–108 °C, the decline amounted to 1% and 60%, respectively. For lignocellulose pellets, temperature was the most important factor for *σ_c_*, followed by the compaction pressure, particles size, and *MC* [19]. On the other hand, the changes of *E_j_* vs. *l_c_* were proportional and had a digressive gradient. It is logical because at the higher *l_c_*, pellets were more densified and higher work was needed in order to the pellets cracking.

The work *E_j_* needed for the pellet to crack increased with an increase in the value of *ρ_p_* because the correlations between these parameters were either high, r = 0.514 for the blend of WS with CS or very high, with r = 0.813 and r = 0.816 for WS and the blend of WS with CC, respectively (Appendix A). High or very high correlations were recorded among *E_j_*, *E*, and *σ_c_*. A negatively high correlation (r ≈ −0.5) was recorded between *E_j_* and *k*; the less the pellet was immersed in water, the more it was resistant to cracking.

In summary, the influence of CC was more significant than CS as an additive to WS due to the need to use larger *E_j_* for the pellets to crack. The involvement of bonding additives did not have a higher influence on the *E_j_*. The influence of the *l_c_* on the resistance of the pellet to cracking during radial compressive load was also confirmed [19]. The influences of *MC* and *t* were contrary and with an exponential shift, with characteristics increasing to a plateau for *E_j_* as the moisture increased and with inverted characteristics as *t* decreased (Figure 1).

The increase in temperature caused changes in the physicochemical properties of the lignocellulosic compounds of WS, leading partially to the decrystallisation of particles. At higher *MC*, this caused a shift of the glass transition temperature towards lower values [27]. The moisture effect vs. temperature was more visible for pure WS. For the WS bonded with CC, it densified at the highest *t* of 108 °C. The characteristics of *E_j_* for the CS and CC additives were different; the former was characterised by concavity with a minimum for the temperature of 108 °C and the latter by convexity with a maximum at 78 °C.

### 3.3. Elasticity Modulus for Pellet Compression E

The average value of *E* was the highest (7.42 MPa) for the pellets made of WS and CS blends and lowest (2.39 MPa) for the pellets made of WS and CC blends. The *E* for the pellets made of pure WS was 4.62 MPa.

The CS has a low porosity crystal structure and is characterised by high adhesiveness and long texture; its addition increased the density effect as indicated in the starch characteristics [28]. Adding CC and improving the plasticity of the blends, including WS, contributed to decreasing the elasticity of pellets.

*E* increased progressively along with *MC*; within the following moisture ranges 10–20% w.b. and 20–30% w.b., *E* growth amounted to 15% and 21%, respectively. WS particles and additives filled out with moisture absorption and could have better elasticity at higher moisture. Moreover, water is incompressible; hence its greater share in the material increases its elasticity. Water absorption by 5–25 μ starch grains occurred at relatively low temperatures [25] and, as a result, *E* was higher within these temperature ranges. The highest *E* (5.83 MPa) was recorded for pellets produced at 93 °C. At 78 °C, the value of *E* decreased by 10% and at the highest temperature (108 °C), it decreased by 40%. This shows that there is optimal moisture at which *E* reaches a maximum value (Figure 2).

The correlation between *E* and *t* was negative and weak. *E* correlated very highly with *E_j_* and *σ_c_*, correlation coefficient values higher for the pellets of WS blends with CC than with CS and similar to the value for pure WS pellets, with an average r value of ~0.7 (Appendix A).

### 3.4. Maximum Tensile Strength σ_c_

The pellets made of WS with CC had the highest average values of *σ_c_* (2.11 MPa). They were higher by 71% than *σ_c_* for the pellets of WS with CS. The highest growth of *σ_c_* was recorded for the increase of *MC* from 10% w.b. to 20% w.b. which was 133%. In the 20–30% moisture range, this growth was 17%. A similar tendency but with a gradient level four times lower was observed for the following die height ranges *l_c_* = 66–76 mm and *l_c_* = 76–86 mm, in which the growth values of *σ_c_* were 28% and 4%, respectively. The reverse tendency in *MC* and *l_c_* was apparent in the temperatures; in the ranges of 78–93 °C and 93–108 °C, the decrease of *σ_c_* amounted to 6% and 63%, respectively. These differences are visible in the *σ_c_* vs. *MC* and *l_c_* diagrams for the three temperature levels (Figure 3). Higher water content decreased the temperature of softening point of lignin and increased the softness of fibrous materials at high temperatures [28]. The highest temperature of 108 °C led to a significant plasticisation of lignin and reduction of pellet strength. Thus, *t* should be lowered as indicated in the production of rice straw pellets, which were unstable at 100 °C and showed proper strength at 60–80 °C [28]. On the other hand, at temperatures above 100 °C, the amount of evaporated moisture increased on the die surface, which hindered the plasticisation of lignin [29].

Many researchers indicate that the durability and strength of pellets or briquettes initially increase with moisture and then decreases with further increases in moisture [19,30,31,32]. This indicates the existence of an optimum moisture value for effective durability and pellet strength during the material compaction process.

The parameters of *σ_c_* correlated highly with *E*, r = 0.515–0.702 but foremost, they were highly correlated with *E_j_*, r = 0.830–0.928 (Appendix A). The higher the *E_j_* and *E*, the more the pellets were characterised by higher *σ_c_*. It is important to note that this high correlation was recorded between independently measured strength parameter values and *ρ_p_.* The parameters of *ρ_p_* are the main indicators of pellet quality, followed by those of *σ_c_*, and pellet water content [33]. The best results were achieved for the pellets produced from the blend of WS with CC for which the correlation between *σ_c_* and the *ρ_p_* was almost perfect with a correlation coefficient of r = 0.904. This result was not consistent, especially without a converse relationship. It was recorded that if *σ_c_* is higher, *ρ_p_* is also higher; however, but the opposite is not always true, as higher *ρ_p_* does not necessarily imply stronger bonds [34]. Comparative studies did not show any relationship between the density and durability of pellets and biomass briquettes [35]. High *ρ_p_* may be caused by the high specific density of the material, which was noted in our research with walnut shells whose particles were characterised by high sphericity and the bonds between them were much weaker [19] than between the fibrous particles that enhanced their strength [32]. Each material requires specific optimal densification conditions [36].

### 3.5. Pellet Water Absorption k

The largest *k* (5.45 g H_2_O·g^−1^ DM) was achieved by crushed pellets made of pure WS. The addition of CC reduced *k* by 12% and the addition of CS by as much as 59%. An increase in the additive ratio from 2% wt/wt to 6% wt/wt increased *k* from 3.56 to 3.69 g H_2_O·g^−1^ DM, by 4% and in a larger additive range (6–10% wt/wt), absorption was reduced from 3.69 to 3.30 g H_2_O·g^−1^ DM; that is, by 34%.

Moisture increased with absorption (Figure 4) but there is no unequivocal tendency between the types of additives. For all bonds, the highest *k* at the lowest *MC* for the CC additive minimal *k* value was achieved at an *MC* of 20–22%. For CS, *k* was the lowest at the highest *MC*.

The pellets made at the lower *l_c_* (66 mm) were characterised by the highest of *k* = 4.00 g H_2_O·g^−1^ DM. Increasing the *l_c_* to 76 mm decreased *k* by 4%. Further increasing the height to 86 mm decreased the water absorption ability of the crushed pellets by another 11%. Thus, the decrease gradient in *k* was progressive.

The temperature at which the pellets were produced induced opposite behaviour. The pellets produced at the lowest temperature of 78 °C had the lowest *k* at 3.35 g H_2_O·g^−1^ DM. An increase in the temperature by 15 °C increased the water saturation of the crushed pellets by 7%. Further increasing the temperature by 15 °C increased *k* by 23%.

In conclusion, it should be stated that the main factors including the type of material, *A*, *MC*, *l_c_*, and *t*, which varied during the production of pellets, had different influences on *k* than on *ρ_p_* and pellets strength. Therefore, the relations among these criterion parameters and *k* were contrary, as evidenced by the high but negative correlation coefficient values (Appendix A). The relation between *k* and *ρ_p_* was the highest because the values of the correlation coefficients r ranged from −0.502 to −0.645. Among the strength parameters, *σ_c_* correlated best with *k*, for which the values of correlation coefficients ranged −0.505 to −0.574. Due to the k criterion, producing pellets with lower density and compressive strength was favourable.

The most critical parameter of *k* was likely the specific surface area, which was larger for the pellets that cracked and were flattened easily under compressive loads. A similar assumption was formulated in the study of moisture absorption by wood chips western red cedar (*Thuja plicata*), Douglas fir (*Pseudotsuga*
*menziesii*), and western juniper (*Juniperus occidentalis*) [3].

In contrast, the water absorption of chopped WS with an *MC* of 10% w.b. was 4.97 g H_2_O·g^−1^ DM. This value is lower than that of pelleted WS but higher than that of the WS blend with CS or CC, 6.27 and 6.43 g H_2_O·g^−1^ DM, respectively. The additive of 2–10% wt/wt was not statistically significant in the differentiation of water absorption because the results were within the narrow range of 6.20–6.51 g H_2_O·g^−1^ DM, composed of a homogenous group. The research results were presented only for the smallest WS moisture as it is reasonable to use dry straw as bedding.

Pelleted WS, however, had better properties in terms of the appearance of FPD, which was smaller than for birds kept on chopped straw [2] because chopped straw has sharp edges [1].

Due to water absorption, CC proved to be a better addition to WS than CS. The addition of CC to the organic substance in the pelleting process facilitates the form and a small addition reduces bedding dusting [37]. The addition of gypsum to the bedding lowered its pH due to the precipitation of CC and bicarbonate [38]. The blends of straw with CC increased N mineralization and promoted NH_4_^+^ formation by increasing urease activity and buffering against large pH increases [39].

In terms of water absorption and compared to CC, the effect of CS as an additive to WS was diverse and without diversification toward additive rations. The main reason for this might be a low compaction pressure, which resulted from low friction between the WS and CS blends and die hole surface. Although the available literature does not provide enough explanations concerning the bond mechanisms of CS with organic substances, there are studies regarding the compaction of blends to pellets or briquette [40,41]. For example, briquettes from a blend of carbonised corncob and CS, corn starch, and gelatine were characterised by favourable density and strength as well as acceptable stability after several months of storage [41].

### 3.6. Optimisation of Pelleting Conditions Due to Pellets Strength and Water Absorption

The objective functions were limited to basic restrictions that were the highest *E_j_*, *E*, *σ_p_*, and *k*. The conclusions from the results of studies and statistical analyses allowed to develop mathematical models for strength parameters and *k*. On the basis of models, the optimal values of variables directing the compression process were set (Table 3). In Table 3, additionally, the parameters for the *ρ_p_* have been included in order to compare these conditions with the conditions characterising pellets strength and *k*. The graphic interpretations of the answers functions were presented in the Figure 1, Figure 2, Figure 3 and Figure 4 by marking on the charts the optimal point or a point with the highest value if the optimum was outside the range of studies. The analysis was conducted separately for WS and its blends with CS and CC and also for all connected data to check the possibility of generalising mathematical models. In Appendix A, only the regressive factors important from the statistical point of view were included and for regression the results of the Fisher-Snedecor *F* test and multiple correlation coefficient values. The objective functions were limited to the basic restrictions which were the highest values of *E_j_*, *E*, *σ_p_*, and *k*.

The analysed objective functions achieved the optimal values at diverse decision-making factors but due to this study’s purpose, *A* was an important value. Interestingly, the optimal CS additive was 4% wt/wt for the *σ_c_* and 6.1% wt/wt for *k*. The optimal CC additive was near zero for all other objective functions. At a generalised approach, the *A* value for objective functions *ρ*_p_, *E_j_*, *E*, and *σ*_c_ was 4% wt/wt.

The best strength pellet parameters of pure WS were achieved in similar pelleting conditions; at moisture 23–30% w.b., *l_c_* = 86 mm, and *t* around 91 °C. These last two variables were the largest for WS compared to the variables of other types of materials.

Comparing the pure WS to the blend of WS with CS, the best pellets in terms of strength were achieved under the conditions of higher moisture 30% w.b., lower *l_c_* of approximately 70 mm, and lower temperature of 80 °C.

Adding CC to WS changed the pelleting conditions mostly with regard to *MC*, which should not be higher than 16% w.b. In these conditions, including *l_c_* of approximately 76 mm and *t* = 78 °C, the pellets with the best strength parameters were achieved. The required *MC* and *t* of the WS and CC blend were the lowest compared to the other two materials. The compaction of the WS and CC blend required lower *l_c_* than during the pelleting of pure WS and was higher than for the blend of WS with CS.

The conditions for pelleting the WS blend with CS were different than for the pure WS and WS blend with CC. Similar characteristics of the parameters for these last two materials affected the generalised pelleting conditions. The results show that pellets with the highest *ρ_p_* (853 kg·m^−3^) and *σ_c_* (4.17 MPa), which required the largest *E_j_* (6.10 mJ·mm^−2^) to crush, were produced from a blend of WS with CC.

Compared to the strength requirements, the criterion of the highest absorption *k* required different pelleting conditions. To achieve the highest *k*, material compression should take place in the conditions of the lowest *MC* (10% w.b.), lowest *l_c_* (66 mm), and highest *t* (108 °C) according to the tested range of control variables. The compression of the WS blend with CS requires a lower *t* of 81 °C; however, in the area of optimal *MC* and *l_c_*, the *k* characteristics were only slightly different.

For the optimal conditions for which *k* was the highest, the strength parameters are unacceptable and even unrealistic for *E_j_*, and *σ_c_*. Therefore, it was assumed that the criterion for choosing pelleting conditions should be based on the pellet strength requirements and *ρ_p_*. It was taken into account that the pellet’s strength parameters, in particular, the *E_j_*, and *σ_c_*, highly correlated with the *ρ_p_*. In addition, the assessment of pelleting was expressed by the criterion of *ρ_p_*. Taking into account previous analyses and explanations, it was assumed that the *ρ_p_* will also be a criterion for determining pelleting conditions in relation to the *E_j_*, *E*, and *σ*_c_. Under these technical conditions, pellets with favourable strength properties were obtained, and *k* was a derivative of these criteria. For optimal pelleting conditions with the addition of 4% wt/wt, *MC* = 23.3% w.b., *l_c_* = 78.5 mm, and *t* = 80.2 °C, the parameter values were calculated based on mathematical models listed in Appendix A. In these technical conditions, pellets were produced whose loaded radial compressive force was characterised by *E_j_* = 3.22 mJ·mm^−2^, *E* = 5.78 MPa, and *σ_c_* of 2.68 MPa, with a *k* value of 2.60 g H_2_O·g^−1^ DM. This *k* value was close to the values reported in the available literature [3]. For this scenario, the calculated parameter values were smaller than the largest values that were obtained from mathematical models for optimal conditions. For these optimal conditions, the objective function values were 3.66 mJ·mm^−2^, 6.77 MPa, 3.06 MPa, and 6.28 g H_2_O·g^−1^ DM, respectively. The relative differences among the calculated values with respect to the most favourable values were 12, 15, 12, and 59%, respectively. Thus, the strength of the pellets produced under optimal conditions due to *ρ_p_* was only slightly lower than the pellets obtained under optimal conditions for each strength parameter separately. For these considered conditions, the differences between the values of *k* were significant. The presented analysis and interpretation of the test results indicate that in the quality assessment of pellets one cannot rely on the criterion of *k*. Pellets produced under such conditions would be unacceptable due to their low density and strength. During transport, storage, and handling, they would crumble and dustiness would increase, implying a lower quality of crushed pellets for bedding.

## 4. Conclusions

The pellets produced from crushed WS and its blends with CS or CC with an *A* of 2–10% wt/wt, with a *MC* of 10–30% w.b., *l_c_* of 66–86 mm, and *t* of 78–108 °C were evaluated on the basis of strength, and after crushing them, they were evaluated on the basis of *k*.

Parameters of *E_j_*, *E*, and *σ_c_* positively and highly correlated between mutual values, and all three strength indicators negatively correlated with *k*.

Based on five control variables, regression models were developed, and target function values were determined with basic constraints indicating optimal pelleting conditions according to the pellet strength and *k* value.

It was assumed that the produced pellets should be most resistant to compressive loads. As the pellet strength parameters correlated highly with the *ρ_p_*, which was the criterion for assessing the pelleting process, to maintain the inference consistency, the density was assumed as a criterion for assessing the pellets.

For optimal conditions of *A* = 4% wt/wt, *MC* = 23% w.b., *l_c_* = 78 mm, and *t* = 80 °C, *E_j_* was 3.22 mJ·mm^−2^, *E* = 5.78 MPa, the compressive stress of pellets was 2.68 MPa and *k* 2.60 g H_2_O·g^−1^ DM. In this scenario, the values of the parameters characterising the quality of the pellets intended for bedding were smaller than the largest calculated based on the optimisation of the objective function of the four criterion parameters; i.e., *E_j_*, *E*, *ρ_c_*, and *k* were smaller by 12%, 15%, 12%, and 59%, respectively.

The pellets produced from the WS blend with the CC additive had better strength parameters, and the crushed pellets had better *k* than those produced using the CS additive.

It is recommended that the pellets from the WS and CC blends be produced under the specified technical conditions and after crushing, evaluated via comparative tests with chopped WS as bedding material.

## Figures and Tables

**Figure 1 materials-13-04375-f001:**
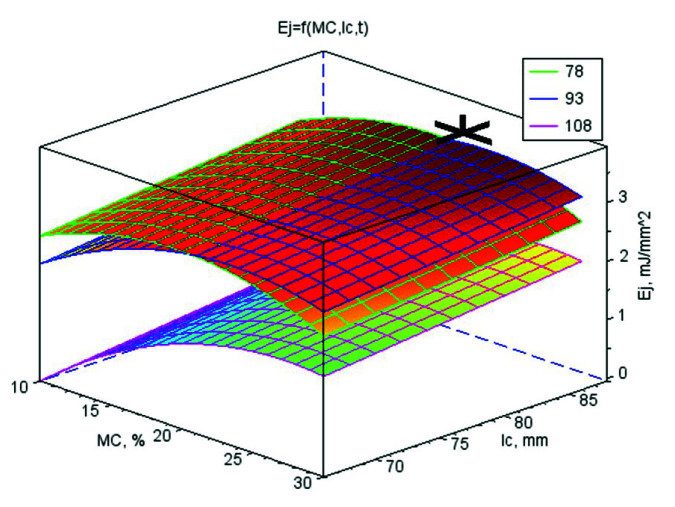
The surface chart of specific pellet compression work *E_j_* vs. material moisture *MC* and die height *l_c_* for three material temperature levels *t* (78, 93, and 108 °C) with the marked optimal point *A* = 0.00% wt/wt, *MC* = 21.6% w.b., *l_c_* = 86 mm, *t* = 86 °C, for which optimum value of *E_j_* is 3.66 mJ·mm^−2^.

**Figure 2 materials-13-04375-f002:**
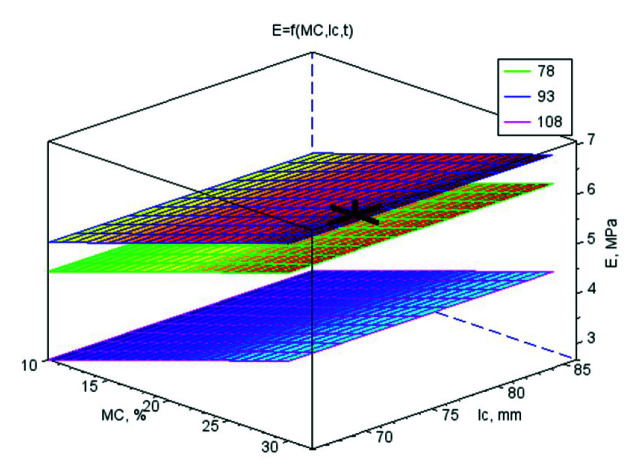
The surface chart of the elasticity modulus for pellet compression *E* vs. material moisture *MC* and die height *l_c_* for three material temperature levels *t* (78, 93 and 108 °C) with the marked highest point *A* = 4.00% wt/wt, *MC* = 30.0% w.b., *l_c_* = 71.0 mm, *t* = 88.5 °C, for which the optimal value of *E* is 6.77 MPa.

**Figure 3 materials-13-04375-f003:**
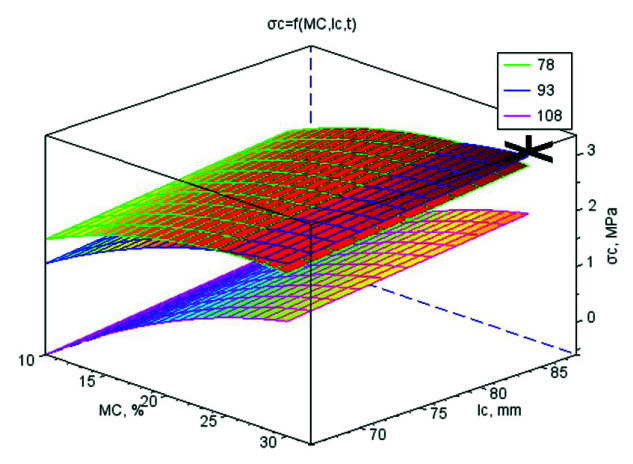
The surface chart of pellet tensile strength *σ*_c_ vs. material moisture *MC* and die height *l_c_* for three material temperature levels *t* (78, 93, and 108 °C) with the marked optimal point *A* = 4.00% wt/wt, *MC* = 30.0% w.b., *l_c_* = 86 mm, *t* = 87.6 °C, for which the optimal value of *σ*_c_ is 3.06 MPa.

**Figure 4 materials-13-04375-f004:**
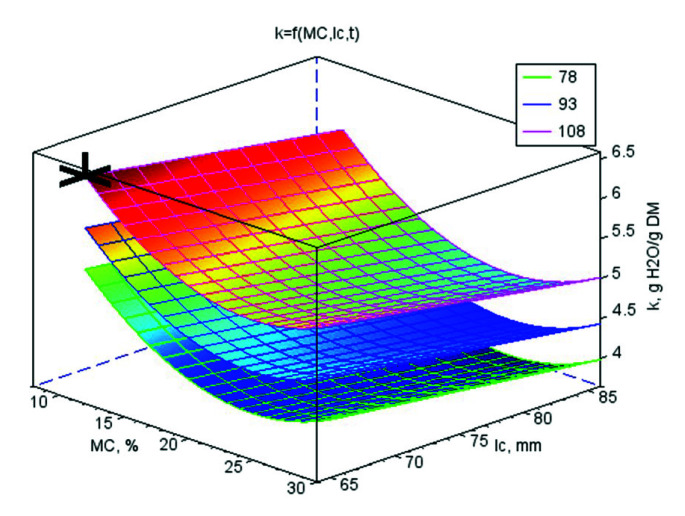
The surface chart of water absorption by crushed pellets *k* vs. material moisture *MC* and die height *l_c_* for three material temperature levels *t* (78, 93, and 108 °C) with marked highest point *A* = 0.00% wt/wt, *MC* = 10.0% w.b., *l_c_* = 66 mm, *t* = 108.0 °C, for which the optimal value of *k* is 6.28 g H_2_O·g^−1^ DM.

**Table 1 materials-13-04375-t001:** Pelleting factors and levels.

Level	Moisture Content, *MC*, % w.b.	Material Temperature, *t*, °C	Die Height, *l_c_*, mm	Additive Ratio, *A*, %	Additive Type
1	10	78	66	0	CS
2	20	93	76	2	CC
3	30	108	86	6	–
4	–	–	–	10	–

**Table 2 materials-13-04375-t002:** Results of analysis of variance, mean values, and standard deviation (SD) of dry matter (DM) single pellet density, specific pellet compression work, elasticity modulus for pellet compression, tensile strength, water absorption by crushed pellets for different type of additive, *A*, *MC*, *l_c_*, and *t*.

Factor	*ρ_p_*, kg m^−3^	*E_j_*, mJ·mm^−2^	*E*, MPa	*σ_c_*, MPa	*k*, g H_2_O·g^−1^ DM
*p*-value					
Additive	0.0285	<0.0001	<0.0001	<0.0001	<0.0001
*A*	<0.0001	0.0523	0.5438	0.2428	<0.0001
*MC*	<0.0001	0.0002	<0.0001	<0.0001	<0.0001
*l_c_*	<0.0001	<0.0001	0.4100	<0.0001	<0.0001
*t*	<0.0001	<0.0001	<0.0001	<0.0001	<0.0001
Mean and ± SD for type of additive					
WS	490 ^c*^ ± 23	2.53 ^b^ ± 0.19	4.62 ^b^ ± 0.30	1.95 ^b^ ± 0.17	5.45 ^c^ ± 0.16
CS	405 ^a^ ± 11	1.28 ^a^ ± 0.08	7.42 ^c^ ± 0.26	1.23 ^a^ ± 0.09	2.24 ^a^ ± 0.04
CC	475 ^b^ ± 18	2.66 ^b^ ± 0.20	2.39 ^a^ ± 0.12	2.11 ^b^ ± 0.15	4.80 ^b^ ± 0.11
Mean and ± SD for additive ratio *A* (% wt/wt)					
0	490 ^c^ ± 23	2.53 ^c^ ± 0.19	4.62 ^a^ ± 0.30	1.95 ^a^ ± 0.17	5.45 ^c^ ± 0.16
2	446 ^b^ ± 18	2.07 ^b^ ± 0.19	5.19 ^a^ ± 0.32	1.75 ^a^ ± 0.15	3.56 ^b^ ± 0.15
6	430 ^a^ ± 18	1.92 ^a^ ± 0.19	4.80 ^a^ ± 0.31	1.68 ^a^ ± 0.15	3.69 ^b^ ± 0.15
10	443 ^ab^ ± 19	1.92 ^a^ ± 0.21	4.73 ^a^ ± 0.32	1.58 ^a^ ± 0.15	3.30 ^a^ ± 0.12
Mean and ± SD for material moisture content *MC* (% w.b.)					
10	372 ^a^ ± 15	1.12 ^a^ ± 0.11	4.12 ^a^ ± 0.24	0.85 ^a^ ± 0.09	4.40 ^c^ ± 0.14
20	495 ^c^ ± 19	2.51 ^b^ ± 0.20	4.73 ^b^ ± 0.25	1.98 ^b^ ± 0.15	3.35 ^a^ ± 0.11
30	479 ^b^ ± 14	2.57 ^b^ ± 0.17	5.71 ^c^ ± 0.32	2.31 ^c^ ± 0.14	3.64 ^b^ ± 0.14
Mean and ± SD for die height *l_c_* (mm)					
66	402 ^a^ ± 15	1.81 ^a^ ± 0.17	4.75 ^a^ ± 0.30	1.43 ^a^ ± 0.13	4.00 ^c^ ± 0.15
76	479 ^c^ ± 17	2.12 ^b^ ± 0.18	4.98 ^a^ ± 0.27	1.83 ^b^ ± 0.14	3.83 ^b^ ± 0.14
86	466 ^b^ ± 17	2.28 ^b^ ± 0.17	4.84 ^a^ ± 0.26	1.90 ^b^ ± 0.14	3.56 ^a^ ± 0.12
Mean and ± SD for material temperature *t* (°C)					
78	542 ^c^ ± 18	2.60 ^b^ ± 0.19	5.25 ^b^ ± 0.28	2.25 ^b^ ± 0.15	3.35 ^a^ ± 0.12
93	494 ^b^ ± 16	2.57 ^b^ ± 0.19	5.83 ^c^ ± 0.27	2.11 ^b^ ± 0.15	3.60 ^b^ ± 0.12
108	310 ^a^ ± 10	1.03 ^a^ ± 0.08	3.48 ^a^ ± 0.25	0.79 ^a^ ± 0.07	4.44 ^c^ ± 0.15

*ρ*_p_, single pellet density; *E_j_*, specific pellet compression work; *E*, elasticity modulus for pellet compression; *σ_c_*, tensile strength; *k*, water absorption by crushed pellets; *A*, additive ratio; *MC*, material moisture contents; *l_c_*, die height; *t*, material temperatures. * different letters in each column and factors within a value represent a significant difference at *p* < 0.05 using Tukey’s test.

**Table 3 materials-13-04375-t003:** The most favourable condition for characteristics parameters of pellets strength and the single pellets density and water absorption by crushed pellets with three different types of materials.

	*ρ_p_*	*E_j_*	*E*	*σ* _c_	*k*
Wheat straw (WS)					
*MC*, %	13.2	30.0	23.2	30.0	10.0
*l_c_*, mm	86.0	86.0	86.0	86.0	66.0
*t*, °C	78.9	91.6	91.2	90.9	108
Optimum value	785 kg·m^−3^	4.64 mJ·mm^−2^	8.69 MPa	3.56 MPa	6.99 g H_2_O·g^−1^ DM
Wheat straw and cassava starch (WS + CS)					
*A*, %	0.00	0.00	0.00	4.00	6.10
*MC*, %	30.0	30.0	30.0	30.0	10.0
*l_c_*, mm	73.3	71.0	66.0	76.1	66.0
*t*, °C	78.0	78.0	87.8	78.0	81.2
Optimum value	682 kg·m^−3^	2.30 mJ·mm^−2^	11.96 MPa	3.16 MPa	3.15 g H_2_O·g^−1^ DM
Wheat straw and calcium carbonate (WS + CC)					
*A*, %	0.00	0.00	0.00	0.00	0.00
*MC*, %	18.1	16.1	10.0	16.9	10.0
*l_c_*, mm	86.0	71.0	86.0	71.0	66.0
*t*, °C	78.0	78.0	78.0	78.0	108.0
Optimum value	853 kg·m^−3^	6.10 mJ·mm^−2^	5.54 MPa	4.17 MPa	6.99 g H_2_O·g^−1^ DM
For all					
*A*, %	4.00	0.00	4.00	4.00	0.00
*MC*, %	23.3	21.6	30.0	30.0	10.0
*l_c_*, mm	78.5	86.0	71.0	86.0	66.0
*t*, °C	80.2	86.0	88.5	87.6	108.0
Optimum value	633 kg·m^−3^	3.66 mJ·mm^−2^	6.77 MPa	3.06 MPa	6.28 g H_2_O·g^−1^ DM

*A*: additive ratio, *MC*: moisture content, *l_c_*: die height, *t*: material temperature, *ρ_p_*: DM single pellet density, *E_j_*: specific pellet compression work, *E*: elasticity modulus for pellet compression, *σ_c_*: pellets tensile strength, *k*: water absorption by crushed pellets.

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
