# Peer review of "Characterisation of Wheat Straw Pellets Individually and in Combination with Cassava Starch or Calcium Carbonate under Various Compaction Conditions: Determination of Pellet Strength and Water Absorption Capacity"

_materials, 2020, doi:10.3390/ma13194375_

Round 1

Reviewer 1 Report

The aim of this study was to determine the most favourable combinations of the density parameters of wheat straw by itself and blended with cassava starch or calcium carbonate as bonding additives by maximising the pellets’ compressive strength and water absorption by crushed pellets. It is an interesting topic. Also, orthogonal arrays with mixed levels are very useful in factorial experiments when factors with different numbers of levels are to be investigated. However, despite the large amount of data produced, the manuscript cannot be accepted in the current form for publication in materials journal.

Specific comments

  1. One limitation associated with mixed-level orthogonal arrays is that many of them require a large number of runs. Authors should define how they constructed the orthogonal array for the mixed level DOE used in the study.
  2. Did authors add center point in the design? If not, this is necessary in order the design to provide a measure of process stability and inherent variability and to check for curvature.
  3. Authors should add the Pareto plot of effects, which will allow the readers to detect the factor and interaction effects that are most important to the process.

Author Response

Reviewer 1

Comments and Suggestions for Authors

The aim of this study was to determine the most favourable combinations of the density parameters of wheat straw by itself and blended with cassava starch or calcium carbonate as bonding additives by maximising the pellets’ compressive strength and water absorption by crushed pellets. It is an interesting topic. Also, orthogonal arrays with mixed levels are very useful in factorial experiments when factors with different numbers of levels are to be investigated. However, despite the large amount of data produced, the manuscript cannot be accepted in the current form for publication in materials journal.

We would like to thank the reviewer very much for reviewing the article and for all the comments and suggestions that contributed to the improvement of the paper. We really appreciate the hard work you have done and want to acknowledge the tremendous contribution. We hope that we have responded to all comments and made all changes accordingly. Thank you very much.

Below are the answers to each comment. In the text, in order to facilitate the revision of the article, all changes were marked in blue.

Specific comments

  1. One limitation associated with mixed-level orthogonal arrays is that many of them require a large number of runs. Authors should define how they constructed the orthogonal array for the mixed level DOE used in the study.

Answer (A): Thank you very much for this comment. The comment is very important indeed. We did the tests according to DOE, but in order to increase the accuracy of the modeling, we finally performed tests for the full set with three replications for each factor level. We were not limited by time and economic limits. An explanation has been added in the second point below.

  1. Did authors add center point in the design? If not, this is necessary in order the design to provide a measure of process stability and inherent variability and to check for curvature.

A: The midpoint of the factor range is the center point. To make it clear, I corrected the title of Table 1. Pelleting factors and levels. I added (Lines 134-135): In order to receive the more reliable database, the tests for each combination of factors and levels were performed.

  1. Authors should add the Pareto plot of effects, which will allow the readers to detect the factor and interaction effects that are most important to the process.

A: Thank you very much for this comment. Adding a Pareto chart would repeat the information already contained in Table S1. Correlation matrix for type of additive, additive ratio (A), material moisture content (MC), die height (lc), material temperature (t), DM single pellet density (ρp), specific pellet compression work (Ej), elasticity modulus for pellet compression (E), pellet tensile strength (σc), water absorption by crushed pellets (k) and for different types of material.

This is obviously a different form but contains the same information. The values of the correlation coefficients indicate which factors affect the measured parameters and indicate the interrelationships between the parameters. Readers can easily detect relationships that are mentioned in the comment.

I would like to emphasize that the scientific goal was not to indicate the hierarchy of factors, but to determine the optimal values of factors - pelletisation conditions, for which the target function reaches the maximum or minimum, using the optimization method. And such conditions of conducting the process were indicated. If pelletisation is carried out under these optimal conditions, the pellet density and pellet strength will be the best, and the water absorption of the crushed pellets will be approved. This is the key inference resulting from our work.

Reviewer 2 Report

The study highlights  the characterisation of wheat straw pellets individually and in combination with cassava starch/calcium carbonate under various compaction conditions in order for determination of pellet strength and water absorption capacity. It is so interesting and resourceful to the future readers and researchers.  Overall, the introduction section is very well written and informative. Materials and methods section is well organized. The same applies to the results and conclusion. Grammar is also correctly used.

Therefore, the paper is considered worthy for publication in “Materials” Journal.

Just some minor comments to improve the quality of the paper:

The literature review can be improved with more relevant literature because, in its current form, only eleven references are used.

Is it possible to show the data about the (chemical,...) properties of the wheat straw pellets individually and in combination with cassava starch/calcium carbonate? If so, please add them in the manuscript to be more informative for the future readers.

Author Response

Reviewer 2

Comments and Suggestions for Authors

The study highlights  the characterisation of wheat straw pellets individually and in combination with cassava starch/calcium carbonate under various compaction conditions in order for determination of pellet strength and water absorption capacity. It is so interesting and resourceful to the future readers and researchers.  Overall, the introduction section is very well written and informative. Materials and methods section is well organized. The same applies to the results and conclusion. Grammar is also correctly used.

Therefore, the paper is considered worthy for publication in “Materials” Journal.

We would like to thank the reviewer very much for reviewing the article and for all the comments and suggestions that contributed to the improvement of the paper. We really appreciate the hard work you have done and want to acknowledge the tremendous contribution. We hope that we have responded to all comments and made all changes accordingly. Thank you very much.

Below are the answers to each comment. In the text, in order to facilitate the revision of the article, all changes were marked in blue.

Just some minor comments to improve the quality of the paper:

The literature review can be improved with more relevant literature because, in its current form, only eleven references are used.

Answer (A): As this article is one of three in the series, I focused on the literature that is directly related to the properties of pellets, mainly the water abortion of pellets. There are only few articles closely related to this issue in the available literature. Originally, the article was more extensive, but due to the journal's requirements, I had to shorten the article. Now I have added a paragraph that was previously deleted (Lines 85-90).

The moisture content of crushed straw was statistically higher than pelleted wheat straw, chipwood, and rice straw [2, 8]. This result may be linked with the higher ability of crushed straw to absorb and release the water.

The comparative studies with pine shavings and sawdust have shown that shredded pellets have a good moisture retention capacity, an acceptable level of fine particles and a low level of chemical contamination [9].

Is it possible to show the data about the (chemical,...) properties of the wheat straw pellets individually and in combination with cassava starch/calcium carbonate? If so, please add them in the manuscript to be more informative for the future readers.

A: Thank you very much for this comment. Due to the purpose of the article, I did not provide the chemical properties of the materials. The article is focused on the physical and mechanical properties of pellets. It is not possible to present all research results in one article. We will use this consideration in writing another article on the thermal properties of these materials. Thank you again for this comment.

Reviewer 3 Report

This paper concerns the characterization and optimization of wheat straw (WS) pellets with additives. This is a potentially interesting paper after the following comments are addressed.

  1. Relevant references need to be cited in the Introduction. For example, the paper states, “Previous considerations that take into account the relationships among parameters …” (line 69-71), but no “previous” papers are cited here.

  1. The relevance of cassava starch (CS) and calcium carbonate (CC) additives needs to be discussed in the Introduction. Questions that need to be answered include why they are selected, whether they were studied previously, etc.

  1. Please justify how each parameter in Table 1 are determined and why these particular values are selected for this paper.

  1. The paper focuses on selected values of several pelleting factors (e.g., Table 1). The applicability of the produced results remains unclear. Please justify and elaborate.

Author Response

Reviewer 3

Comments and Suggestions for Authors

This paper concerns the characterization and optimization of wheat straw (WS) pellets with additives. This is a potentially interesting paper after the following comments are addressed.

We would like to thank the reviewer very much for reviewing the article and for all the comments and suggestions that contributed to the improvement of the paper. We really appreciate the hard work you have done and want to acknowledge the tremendous contribution. We hope that we have responded to all comments and made all changes accordingly. Thank you very much.

Below are the answers to each comment. In the text, in order to facilitate the revision of the article, all changes were marked in blue.

  1. Relevant references need to be cited in the Introduction. For example, the paper states, “Previous considerations that take into account the relationships among parameters …” (line 69-71), but no “previous” papers are cited here.

Answer (A): Thank you for pointing out the logical error. In the line 71, I added two references: Makovskis et al., 2016, Kheravii et al., 2017 – [2,5]

  1. The relevance of cassava starch (CS) and calcium carbonate (CC) additives needs to be discussed in the Introduction. Questions that need to be answered include why they are selected, whether they were studied previously, etc.

A: Thank you. This is a fair point in regard to this article, but since this article is one of three that constitute a PhD thesis, I did not want to repeat the information. As previous articles are already published, I referred to those articles. I wanted to avoid the situation where I justify my choice of additives and factors in each article. Then the reviewers of the doctoral dissertation in the form of these three articles, would indicate errors related to the repetition of information.

In lines 120-121 I added a sentence: The selection of additives and factors with their level of values were presented in our two previous articles [14,15].

  1. Please justify how each parameter in Table 1 are determined and why these particular values are selected for this paper.

A: Thank you very much for paying attention to this aspect. As I wrote earlier, the selection of factors and their value levels were justified in the first article of this series. I would like to explain that I had to shorten the article by over 2,000 words, because it was too long and did not meet the journal's requirements.

I added an explanation (Lines 127-131).

The additive ratio of 0% wt/wt means pure WS. These described factors are the most important characteristics of the material and the pelleting process. The range of values for these factors was selected on the basis of preliminary research with taking into account the practical use. More information about the selection of factors were provided in our first article [15].

  1. The paper focuses on selected values of several pelleting factors (e.g., Table 1). The applicability of the produced results remains unclear. Please justify and elaborate.

A: Thank you for your attention. It is similar to the previous one. For a better grasp and understanding, I have added sentences (Lines 131-135). The selection of additives was dictated by their physicochemical properties, hygroscopicity, neutrality towards the environment and the positive effect on the litter properties. A wide range of factor values will allow for more reliable determination of optimal pelleting conditions, due to the strength of the pellets and their water absorption capacity. This is the basis for the applicability of the obtained results in practice.

Round 2

Reviewer 1 Report

The manuscript was imporved according to Reviewer's comments and thus it is acceptable for publication in Materials in the present form.

Reviewer 3 Report

The authors have addressed all reviewer comments. The paper can be now accepted.